# Endocrine Disruptors and Endometrial Cancer: Molecular Mechanisms of Action and Clinical Implications, a Systematic Review

**DOI:** 10.3390/ijms23062956

**Published:** 2022-03-09

**Authors:** Donatella Caserta, Maria Paola De Marco, Aris Raad Besharat, Flavia Costanzi

**Affiliations:** Department of Medical and Surgical Sciences and Translational Medicine, Sapienza University of Rome, Sant’Andrea University Hospital, Via di Grottarossa 1035, 00189 Rome, Italy; donatella.caserta@uniroma1.it (D.C.); demarco.mariapaola@gmail.com (M.P.D.M.); besharataris@gmail.com (A.R.B.)

**Keywords:** endometrial cancer, endocrine disruptors, bisphenol A, Polycyclic aromatic, Flame Retards, Organoclorurate, Alkylphenol Ethoxylates, mycotoxins, cadmium

## Abstract

It has been widely demonstrated that endocrine disruptors play a central role in various physiopathological processes of human health. In the literature, various carcinogenic processes have been associated with endocrine disruptors. A review of the molecular mechanisms underlying the interaction between endocrine disruptors and the endometrial cancer has been poorly developed. A systematic review was performed using PubMed^®^/MEDLINE. A total of 25 in vivo and in vitro works were selected. Numerous endocrine disruptors were analyzed. The most relevant results showed how Bisphenol A (BPA) interacts with the carcinogenesis process on several levels. It has been demonstrated how BPA can interact with hormonal receptors and with different transcription proliferative and antiproliferative factors. Furthermore, the effect of Polycyclic aromatic hydrocarbons on Aryl hydrocarbon receptors was investigated, and the role of flame retardants in promoting proliferation and metastasis was confirmed. The results obtained demonstrate how the mechanisms of action of endocrine disruptors are manifold in the pathophysiology of endometrial cancer, acting on different levels of the cancerogenesis process.

## 1. Introduction

Endometrial cancer (EC) is the sixth most common cancer in women; Four hundred seventeen thousand cases and 97,000 deaths were recorded in 2020 [1]. About 80% of these lesions are associated with a state of hyperestrogenism that plays a fundamental role and characterizes endometrioid-type neoplasms, which arise at the end of a carcinogenic pathway, including a series of precancerous lesions (complex hyperplasia, possibly with atypia) [2]. The main risk factors of these forms are represented by: high number of anovulatory cycles, nulliparity, late menopause, obesity, diabetes and arterial hypertension. An additional risk factor may be estrogen hormone therapy (not associated with progestins) and a paradoxical effect of hormone therapy with estrogen antagonists used to treat breast cancer [3].

On the other hand, estrogen-independent endometrial neoplasms are associated with poor differentiation, arise later (menopause), have severe or clear cell differentiation, and are not associated with previous hyperplasia or dysplasia endometrium [4]. A strong association between cases of ovarian germ cell tumours (OGCT) and endometriosis has been proven [5]. The presence of atypical postmenopausal blood loss or metrorrhagia, intermenstrual bleeding in the peri and premenopausal period, endopelvic tension accompanied by hypogastric and lumbosacral pain and leucoxantorrhea is strongly suspected in EC. An EC should always be suspected in obese or oligoanovulatory women who present frequent and prolonged metrorrhagia episodes in the perimenopausal or premenopausal age. At the same time, the finding of lymphedema in the lower limbs and oedema in the external genitalia must lead to a manifest clinical picture of iliac lymph node diffusion—obturation [6]. Endocrine disruptors (EDCs) are a broad category of molecules or mixtures of substances that alter the normal hormonal function of the endocrine system, causing adverse effects on the health of an organism [7]. The consequences can lead to tumours, (teratogenic) congenital disabilities, impaired reproductive capacities and other developmental disorders concerning the apparatus targeted by the individual interferers. Blocking and up/down-regulation of hormone secretion [8] has been shown that EDCs hurt various aspects of women’s health, particularly fertility, endometriosis, endometrial and breast cancer [9]. Regarding EC, a recent review [10] reported that there is a plausible association between EDCs and EC, but that knowledge of the underlying molecular mechanisms is nevertheless mandatory. In consideration of this analysis and the high prevalence of estrogen-dependent forms in this systematic review, we analyze the association of the action of EDCs with EC from a molecular and clinical point of view. The systematic review methods, carried out following the Preferred Reporting Items for Systematic Reviews and Meta-Analyzes (PRISMA) guidelines, and the PRISMA flow chart (Appendix A) are present in the Appendix A.

## 2. Discussion

### 2.1. EC and BPA

BPA is a synthetic chemical substance produced by the reaction between two phenols and a molecule of acetone. BPA is a chemical mainly associated with other chemicals to make plastics and resins. For example, BPA is used in polycarbonate plastics, rigid, transparent and high-performance plastic. Polycarbonate is used to produce food-grade containers such as returnable beverage bottles, plastic tableware (plates and cups) and food storage containers. BPA is also used to make epoxy resins which are applied to produce films and coatings for cans and vats for drinks and food. BPA can migrate in small amounts into foods and beverages stored in materials that contain BPA. The effect of BPA on women’s health has been extensively studied and investigated [11]; however, specific data on the molecular impact of BPA on EC remains unclear. Recent studies have shown how BPA can favour action in the onset of EC. The molecular mechanisms investigated are different.

A recent murine study [12] has shown, employing a transcriptomic analysis, how a low BPA doses administration (25 or 250 microgr/kg BW/dBPA) for a prolonged period (365 days) alters the estrogenic cycle and is implicated in uterine pathology in a way comparable to estrogenic therapies. In addition, the transcriptomic analysis showed that a common expression of 476 human orthologous genes predicting overall survival (*p* = 1.68 × 10^−5^, hazard ratio = 2.62) is present in patients with EC. Neff et al. [13] also demonstrated, in another murine study, how the chronic oral administration of low doses of BPA alters E and P signals by activating the fibroblast growth factor receptor (FGFr) pathway and phosphorylation of ERK1/2 mitogen-activated protein kinases by inhibiting the transcription of the antiproliferative factor HAND2. These results may suggest an interaction in the process of hyperplasia and carcinogenesis.

Yaguchi [14], through an in vitro analysis, focused his attention on the role of BPA as a promoter of EC cells proliferation. The first impressive result obtained demonstrates that a significant increase in Ki67 activation (*p* < 0.01) is present in the grade I EC cells (Ishikawa cells, HEC265 cells) group treated with BPA and not insignificant grades. The mechanisms underlying this cell proliferation were examined in the study. For both cell lines, it was shown that there is an increased nuclear translocation of estrogen-related receptor γ (ERRγ), which, however, occurs differently in the two cell lines. In particular, Ishikawa cells under the stimulus of BPA determine the secretion, at the level of the extracellular space, of epidermal growth factor (EGF) in a Ca^2+^ dependent manner, resulting in the activation of the EGFR/ERK signalling pathway, which is decisive in cell proliferation. On the other hand, HEC265 cells determine the synthesis of de novo proteins in an independent Ca^2+^ way.

A further study showed that BPA determines a dose-dependent manner, overexpression of the epithelial-mesenchymal transition (EMT) and cyclooxygenase-2 (COX-2) genes [15]. In addition, Chou et al. [16] demonstrate how an epigenetic action of BPA is possible in the carcinogenesis of EC. They report that in the EC, the altered regulation of different MicroRNAs (miRNAs) are evident (reduction of the expression of miR-149 with adverse action on the cell cycle and increase in the expression of miR-107, which activates different mechanisms associated with the proliferation process cellularly). The different expression of miRNAs in endometriosis and ovarian cancer has been studied in several works. Notably, miR-325 and miR-492 were overexpressed in both diseases. The upregulation of miR-325 expression was more critical in ovarian cancer. This result suggests that miR-325 may play a role in transitioning from endometriosis to ovarian cancer [17].

### 2.2. EC and Polycyclic Aromatic

Polycyclic aromatic hydrocarbons (PAHs) belong to a large group of organic compounds, mostly non-volatile, which in indoor air are partly in the vapour phase and partly adsorbed on a particle. The main sources are combustion sources such as kerosene boilers, wood-burning fireplaces and cigarette smoke. PAHs have known negative effects on the environment, on human and animal health, such as evident toxicity for some aquatic organisms and birds, chronic high toxicity for aquatic life, contamination of crops [18]. Several PAHs have been classified by IARC (1987) as “probable” or “possible human carcinogens”, while benzo(a)pyrene (BaP) has recently (2008) been reclassified in group 1 as a “carcinogen for humans” [19]. PAHs commonly present in environmental matrices include benzo(a)pyrene, benzo(b)fluoranthene, benzo(k)fluoranthene, indeno(1,2,3-c,d)pyrene, benzo(a)anthracene, benzo(j)fluoranthene and dibenzo(a,h)anthracene.

The Aryl hydrocarbon receptor (AhR) is a protein that in humans is encoded by the AHR gene and its receptor is a transcription factor that regulates gene expression [20]. Several studies have focused their attention on the role of AhR agonists. Wormke et al. [21] demonstrated that the endometrial activation of the AhR pathway is present using polyhalogenated aromatic hydrocarbons. Furthermore, AhR agonists such as [3H] 2,3,7,8-tetrachlorodibenzo-p- dioxin (TCDD), cause a reduction in the estrogenic response (E2). The molecular mechanisms underlying the activation of AhR in the endometrium were further investigated by Willing et al. [22], who confirmed with the use of polyhalogenated aromatic hydrocarbon agonists the activation of the pathway associated with AhR. It is also shown that TCDD has no anti-estrogenic action in the human telomerase reverse transcriptase-endometrial epithelial cell (hTERT-EEC) and that PCB has an independent AhR pro-migration action. Chen et al. 2017 [23] more recently demonstrated the role of the androgen receptor (AR) poly-glutamine polymorphism (AR-Q) in EC, considering a possible interaction between ARQ polymorphism and toxic environmental agents such as BaP.

### 2.3. EC and Flame Retards

Halogenated flame retardants (HFRs), widely introduced in the early 1970s, are objects or quickly flammable materials commonly used to reduce the development of smoke in the event of a fire and contain the spread of flame. Many of these compounds are associated with adverse health effects, including EDCs, cancer, immunotoxicity, reproductive toxicity and adverse effects on fetal and infant neurological function [24]. Even today, halogenated organic flame retardants account for 25% by volume in terms of world production. Chlorine and bromine are the most widely used halogens [25]. Some flame retardants have been banned and replaced with new compounds with similar structures. The physicochemical properties of polychlorinated biphenyls (PCBs) and their low production costs have facilitated their use as flame retardants. Among the brominated flame retardants (BFRs), polybromobiphenyl ethers (PBDE) were the most used. In 2009, the Stockholm Convention banned their production and use [26]. The so-called “legacy BFRs” are defined as obsolete, but they are still widely monitored due to their ubiquity in the environment. Zhang et al. [27] evaluated the role in EC. They employed an in vitro and in vivo analysis of the effects of 2,2′, 4,4′-tetrabromo diphenyl ether (BDE-47), one of the most frequent polybrominated diphenyl ethers (PBDEs) found in environmental and biological samples. The results obtained have shown an increased ability to metastasize and proliferate when BDE-47 is used. This result shows how even flame retardants can have a pro-carcinogenic action in EC.

### 2.4. EC and Organoclorurate

The acronym PCB stands for Polychlorinated Biphenyls, Polychlorinated Biphenyls, a group of organochlorine industrial chemicals that became a major environmental problem in the 1980s–1990s. These compounds find a wide variety of applications in our society due to their properties. These products become a significant source of pollution due to their indiscriminate use [28].

Similar to many other organochlorines, these substances are very persistent in the environment and bioaccumulate in living systems. Both for their toxicity and their furan contaminants, PCBs present in the environment have become a cause of great concern for their potential impact on human health, especially on human growth and development [29]. Numerous studies have analyzed the action of organochlorines in EC. A case–control study demonstrated an increased risk of EC associated with a concentration of p,p’-dichlorodiphenyldichloroethylene in the adipose tissue (p,p’-DDE) [30]. However, this clinical finding is not confirmed by Weiderpass et al. [31] who, in a case–control study on the blood concentration of 10 PCBs in 154 patients with EC, did not find a significant association with the risk of cancer. According to Weiderpass, Donat-Vargas [32], in an observational study on 36,777 patients, lasting 14 years, the risk of cancer associated with PCB exposure, employing a food intake analysis, is not statistically significant 1.21 (95% CI: 0.73, 2.01). Nevertheless, observing these clinical data, the molecular mechanisms are still unclear.

An in vitro study analyzed the effect of 3,30,4,40,5-Pentachlorobiphenyl (CB126) and 2,20,4,40,5,50-hexachlorobiphenyl (CB153) on the endometrium. The study demonstrated increased cell proliferation and a decrease in Interleukin-8 by CB 126, resulting in the enzymatic activation of SOD1. Thus, the PCBs demonstrate an interaction of its inflammation factors through the ER and AhR receptors [33]. These data suggests that there may be an interaction in the process of carcinogenesis of EC from a molecular point of view.

### 2.5. EC and Alkylphenol Ethoxylates

Ethoxylated alkylene veins (generically called APEOS) constitute a vast category of non-ionic surfactants, characterized by excellent performance, both as detergents and as emulsifiers and dispersants. APEOS are also used in plastic additives and pesticides [34,35,36]. The APEOS biodegradation process generates more toxic products, namely alkylphenols (AP) and nonylphenol (NP) and oxalphenol (OP) [37].

Ethoxylated nonylphenols have been used as surfactants, emulsifiers, dispersants and soakers in various industrial applications (especially in the textile sector) and in consumer products. Other sectors in which alkylphenols are still used are leather treatment and cosmetics and, more generally, in the personal care sector. There is also the use of nonylphenol derivatives as antioxidants in some types of plastics.

As for octylphenol ethoxylates, although less reliable data are available, it seems that they are used in a range of applications similar to that of nonylphenols. For both groups, the changes that have occurred over the past five years in using these compounds are not well documented.

The ingestion of contaminated food or water may result in bioaccumulation of degraded metabolites of APEOS [35,38] and skin inhalation or absorption of APs [39,40].

With their estrogenic potential, it has been shown in mouse studies that NP and OP interfere with estrogenic activity with effects on the uterus resulting in endometrial proliferation [27,41].

In their case–control study, Hui-Ju Wen [42] among women diagnosed with EC (n 49) or uterine leiomyoma (n 247) and healthy women (n 101) performed between 2011 and 2014 evaluated urinary concentrations of OP and NP with Gas chromatography/mass spectrometry. Women with EC reported a significantly higher NP concentration than those in the control group, and the OP concentration was found to be higher in EC patients than in those with LM and controls. Avoid, or at least minimize, exposure to alkylphenols in daily life would seem helpful to improve gynaecological health. However, further investigation is needed as this is a single study with limitations such as the age difference between the groups or the urinary dose alone, making it challenging to observe long-term exposure having a limited urinary half-life.

### 2.6. EC and Mycotoxins

Many cereal products such as beer are frequently contaminated with EDCs such as mycotoxins, which are toxic secondary metabolites of mold. The most famous mycotoxins is zearalenone (ZEN) [43] that mimics the structure of the natural hormone 17b-estradiol and can bind and activate estrogen receptors (ER), impairing fertility, sexual development and the proliferation of hormone-sensitive tissues.

The secondary metabolite produced by Fusarium molds, which often infests corn and other grains [44] has been found in animal feed [45], cereals, baked goods or beer [46].

Metabolites produced by the intestinal microbiota and the human liver are different, including a-zearalenol (a-ZEL) which assumes great relevance for a significantly greater estrogenic potency than ZEN [47]. These compounds could have relevance in the development of oromonosensitive cancer cells such as those of the endometrium.

The study of Pajewska et al. [48], analyzed compounds resulting from the metabolization of ZEN produced by different species of Fusarium directly on surgically excised endometrial tumor tissue.

A total of 61 tissue samples obtained from patients were analyzed between 45 and 88 years of age, 49 with EC and 12 with endometrial hyperplasia. ZEA and its metabolite α-ZEL were not detected by High-performance liquid chromatography with fluorescence detection (HPLC-FLD) in any of the samples with endometrial hyperplasia. The concentration of ZEA reached 44.7 ng/g in HPLC-FLD, confirmed by UHPLC-MS/QTOF analysis of a tissue sample with Endometrioid adenocarcinoma G1. Furthermore, α- ZEL was detected in 47 tissue samples and ZEA was identified in 30 samples analyzed by UHPLC-MS/QTOF. The detection frequency (%) of both analytes differed significantly (Mann–Whitney U test, *p* < 0.01) between tissues with endometrioid adenocarcinoma and endometrial hyperplasia [48].

Another potent phytoestrogen is 8-prenylnaringenin (8-PN) which comes from xanthohumol (XAN) contained in hops for example [49,50]. In beer, we can find mixtures of these EDCs. In the study by Aichinger et al. [51], the interaction of these xenoestrogens was studied, resulting in the combined behavior different from the isolated one.

It has already been shown that genistein soy flavonoids potentiate the estrogenic effects of ZEN on Ishikawa cell culture. Interactions with 8-PN were studied, and unexpectedly, current in vitro data indicate that hop flavonoids help protect against adverse estrogenic effects. In particular, XAN seems to act as a potent antagonist of mycotoxin-induced estrogenicity, significantly suppressing the impact that induces AlP of both ZEN and a-ZEL at nanomolar concentrations. Additionally, 8-PN antagonized the estrogenic stimulus of the two fungal metabolites, although less pronounced.

### 2.7. EC and Cadmium

Cereals, vegetables and tobacco contain cadmium, a carcinogenic heavy metal, resulting from waste from industrial and agricultural activities that release cadmium into the environment [52,53]. This leads to chronic exposure for humans, albeit at low levels. Low iron stores increase their absorption through food, resulting in higher levels in women [54] with health consequences.

Cadmium acts on estrogen signalling and causes inflammation, oxidative stress and alters DNA methylation [55,56,57], relevant for the oncogenic mechanism of hormone-dependent tumours. It can also interfere with the process of clotting and fibrinolysis. Cadmium exposure may be one of the factors inducing these changes in women with EC [58].

Some prospective studies have shown an association between estimated diet cadmium and EC [59].

Another study conducted in 2014 [60] looked at the association between dietary cadmium intake and the risk of these cancers in 155,069 postmenopausal women aged 50 to 79 years.

A total of 1198 EC have been reported. However, no statistically significant associations were observed between dietary cadmium and EC risk after adjustment for potential confounding factors.

### 2.8. EC and and Other EDCs

Methoxychlor (Mxc), an antiparasitic with action on fertility with an uncertain role in carcinogenic processes, has been shown not to have a procarcinogenic action in EC [61].

The same applies to the study conducted on BPA, phthalates and triclosan, in which no correlations were found with the development of endometrial malignancy, although the concentrations studied were thought to have been underestimated [62].

The concentrations of parabens (methylparaben, ethylparaben, N-propylparaben, benzylparaben, isobutylparaben + N-butylparaben) in endometrial and myometrial tissue samples of patients with EC were significantly higher than those of patients with benign disease without a correlation between the concentration and diameter of the tumor, invasion of the myometrium, architectural grade, nuclear grade, invasion of the lymphovascular space and stage of the tumor.

Paraben molecules N-Propylparaben and isobutyl + N-butylparaben were the most frequently detected [63].

The effects of skin care products that may contain some EDCs were also evaluated but no correlation with EC was found in the Norwegian women’s court of 106,978 participants [64]. Some environmental contaminants such as 2,3,7,8-tetrachlorodibenzo-p-dioxin (TCDD) act on estrogen-dependent signalling. These contaminants act through the repression of the AhR function in a dioxin-dependent manner and activation/co-repression for Aryl Hydrocarbon Receptor Nuclear Translocator (ARNT) complex in a dioxin-independent manner [65].

### 2.9. Discussion

The environment has a significant biological impact on human health and disease. For decades, the long-term and persistent effects of the chemical compounds humans are exposed to daily have been ignored due to a lack of scientific information and public awareness.

Observing the analyzed data, in EC the main criticalities encountered in the study of EDCs are heterogeneous (Figure 1). The presence of delayed effects appears evident. In many cases, contact with EDCs does not have immediate effects. Exposure before puberty or during intrauterine life can, for example, lead to significant effects on fertility after many years [66]. The foundations of many adult pathologies could be traced back to exposure to EDCs during life in utero. The temporal gap can be 20–30 years as documented for some neoplasms (cervical and testicular cancer) or even higher as postulated for some dementias [67,68]. A second important issue is that the exposure is often multiple. Most studies on EDCs aim to identify the effects and mechanisms of action of individual substances. However, given the heterogeneous nature of EDCs, it is reasonable to assume that their distribution is ubiquitous and that simultaneous exposure to multiple substances is anything but theoretical. The possible interactions between different substances are largely ignored: it is possible that the mechanisms of action of several EDCs are additive or even synergistic, leading to more striking manifestations in the face of less exposure to the individual components effect measurement. The heterogeneity of EDCs makes it challenging to estimate the exact effects, even in the presence of clinical manifestations attributable to them. In addition, long-term exposure to low doses of EDCs often makes it challenging to identify a causal link between the “culprit” agents and the clinic. Each subject has a unique pattern of exposure to known and unknown EDCs, and the physiological differences in terms of metabolism and body composition contribute to altering their half-life and biological effects. Classical toxicology methods provide only a very partial representation of the effects of EDCs, as do many animal studies.

## 3. Conclusions

In conclusion, the results obtained show how the different EDCs act on different levels of EC pathophysiology, performing on different carcinogenic mechanisms. However, it is easy to define the pathogenesis modalities of EDCs, but it is not easy to define the dosages to which it is dangerous to expose oneself. It is impossible to define each patient’s exposure rates at the tissue level; however, mainly through in vitro cell culture studies, it has been possible for EDCs to detect pro-inflammatory mechanisms, to define oxidative stress, genetic and epigenetic conditions, and have a hormone-sensitive action [69,70,71,72]. In the genetic field, in particular, mutations that activate PIK3CA associated with the loss of ARID1A expression appear to be necessary for cancer development. Conversely, the inactivation of ARID1A alone is not sufficient to cause tumour development [73]. In fact, by imitating the estrogen molecule, they can activate the endometrial receptors, stimulating the proliferation and transformation of hormone-sensitive tissues in a tumour sense and making lifestyle increasingly crucial in preventing cancer, especially the endometrium. However, many studies are still preliminary and developed in vitro, so this review should be a starting point to deepen the studies on EDCs and EC to identify more weapons in fighting and preventing this pathology.

## Figures and Tables

**Figure 1 ijms-23-02956-f001:**
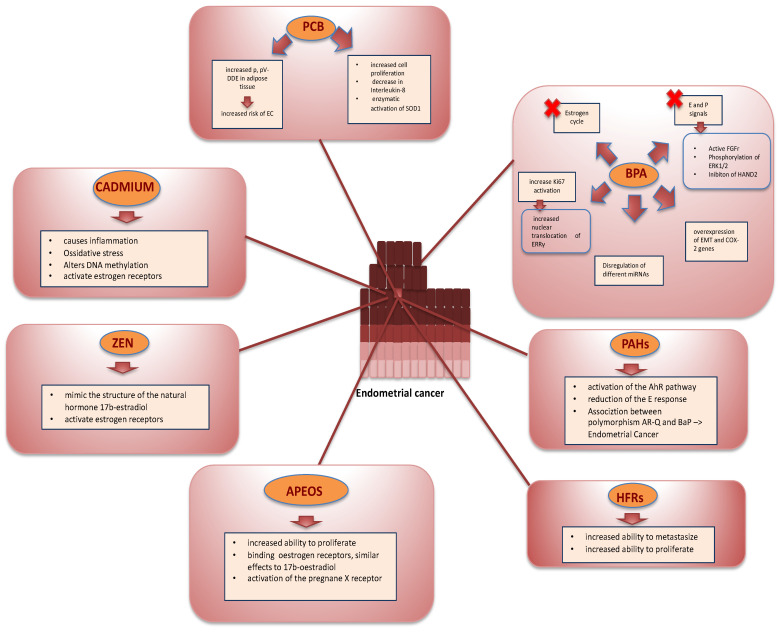
EDCs and EC. Bisphenol A:BPA; E: Estrogen; P:progesteron; fibroblast growth factor receptor: FGFr; Extracellular signal-regulated protein kinases 1 and 2: ERK1/2; Heart- and neural crest derivatives-expressed protein 2:HAND2; epithelial-mesenchymal transition: EMT; cyclooxygenase-2:COX-2; MicroRNAs:miRNAs; increased nuclear translocation of estrogen-related receptor γ (ERRγ); Polycyclic aromatic hydrocarbons: PAHs; aryl hydrocarbon receptor:AhR; poly-glutamine polymorphism: AR-Q; benzo(a)pyrene: BaP; Halogenated flame retardants:HFRs; Ethoxylated alkylene: APEOS; zearalenone:ZEN; Polychlorinated Biphenyls: PCB; p,p’-dichlorodiphenyldichloroethylene: p,p’V-DDE: superoxide dismutase 1:SOD1; red cross: iterferes.

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
