# Peer review of "Endocrine Disruptors and Endometrial Cancer: Molecular Mechanisms of Action and Clinical Implications, a Systematic Review"

_ijms, 2022, doi:10.3390/ijms23062956_

Round 1

Reviewer 1 Report

The review from Caserta and coworkers represent a well-organized systematic review on the topic proposed. There are few minor concerns before its publication listed below:

  • Although the literature was revised according to the PRISMA guidelines, I recommend to eliminate the paragraph “method” with the Figure 1 and eventually include them as supplementary material.
  • I suggest to change the structure and the title of the paragraph as for a classical review. Indeed, I suggest to remove the paragraph "method" (see previous point), change the title to the “result and discussion” and “discussion” paragraphs (the sub-paragraph titles are ok). It is fine to me to let the paragraphs “introduction” and “conclusion” as such.
  • I will suggest to revise the use of starting a new line in several part of the manuscript (e.g. line 179-180).
  • Please revise the sentences at line 237 (“A study” vs “The study of” or “A specific study from”); line 262 (“by soil and soil, resulting from waste from”?), line 306 (“taking EDCs”?). Line 297 and 298 have no sense, please revise.
  • Please revise figure 2, what is the meaning of the red cross?, it is not written in the capture. Could be interesting to include also a table resuming the literature results.
  • Line 240-248, the reference is missing.

Author Response

RESPONSE TO REVIEWERS

REVIEWER N.1

The review from Caserta and coworkers represent a well-organized systematic review on the topic proposed. There are few minor concerns before its publication listed below:

  • Q: Although the literature was revised according to the PRISMA guidelines, I recommend to eliminate the paragraph “method” with the Figure 1 and eventually include them as supplementary material.I suggest to change the structure and the title of the paragraph as for a classical review. Indeed, I suggest to remove the paragraph "method" (see previous point), change the title to the “result and discussion” and “discussion” paragraphs (the sub-paragraph titles are ok). It is fine to me to let the paragraphs “introduction” and “conclusion” as such.

A: Thank you for this comment. The paragraphs have been changed as suggested.

  • Q: I will suggest to revise the use of starting a new line in several part of the manuscript (e.g. line 179-180).

A: Thank you. The text has been revised.

  • Q:Please revise the sentences at line 237 (“A study” vs “The study of” or “A specific study from”)

A: Thank you. The text has been revised.

  • line 262 (“by soil and soil, resulting from waste from”?), line 306 (“taking EDCs”?). Line 297 and 298 have no sense, please revise.

A: Thank you. The text has been revised.

  • Q:Please revise figure 2, what is the meaning of the red cross?, it is not written in the capture.

A: The figures have been revised.

  • Q:Could be interesting to include also a table resuming the literature results.

A:Thank you for this comment. We believe that, although schematic, Figure 2 may be useful for those readers who are not confident or knowledgeable about the subject of the review. Its role is to be figuratively introductory to the topic.

  • Q:Line 240-248, the reference is missing.

A: The reference has been entered

Reviewer 2 Report

Please correct the english to make the article readable. As it is, I could not begin to do the peer review.

Author Response

REVIEWER N.2

Q: Please correct the english to make the article readable. As it is, I could not begin to do the peer review.

 A: The revised version of the manuscript was further reviewed for English language by an English speaking native person.

Reviewer 3 Report

In this article, the authors analyze the association of the action of EDCs with EC from a molecular and clinical point of view. The manuscript is straightforward, well written, and concise and has clear result within the scope of a review article. Definitely deserves to be published and is a valuable contribution to the “International Journal of Molecular Sciences”. Some minor comments need to be addressed before publication.

Minor points:

[1] “1. Introduction”, Lines 36-38:

On the other hand, estrogen-independent endometrial neoplasms are associated with poor differentiation, arise later (menopause), have severe or clear cell differentiation, and are not associated with previous hyperplasia or dysplasia endometrium [4].”.

At that point, the authors should report whether there is an association between endometriosis and the development of endometrioid and clear cell carcinomas. An interesting paper reviewed cases of ovarian germ cell tumors (OGCT) in postmenopausal patients. Among 35 OGCT reported cases, endometrioid carcinoma was the most common epithelial component, and 6 of these events were associated with an endometriotic cyst. In terms of clear cell carcinoma, 1 out of 3 cases had a history of endometriosis as well.

Recommended reference: Boussios S, et al. Malignant Ovarian Germ Cell Tumors in Postmenopausal Patients: The Royal Marsden Experience and Literature Review. Anticancer Res. 2015;35(12):6713-22.

[2] “3.1. EC AND BPA”, Lines 116-119:

They report that in the EC, altered regulation of different MicroRNAs (miRNAs) are evident (reduction of the expression of miR-149 with adverse action on the cell cycle and increase of the expression of miR-107, which activates different mechanisms associated with the proliferation process cellular).”.

There are several studies where differential expression of miRNAs has been studied in either endometriosis or ovarian cancer. Several differentially expressed miRNA in endometriosis compared to ovarian cancer have been found, mainly linked with epithelial–mesenchymal transition. Two common miRNAs overexpressed in both diseases were miR-325 and miR-492. While the expression of miR-325 was upregulated in both diseases, this was more prominent in ovarian cancer, suggesting that miR-325 could have a role in the transition from endometriosis to ovarian cancer.

Recommended reference: Braicu OL, et al. miRNA expression profiling in formalin-fixed paraffin-embedded endometriosis and ovarian cancer samples. Onco Targets Ther. 2017 Aug 28;10:4225-4238.

[3]4. Conclusion”, Lines 341-343:

It has been possible for EDCs to pronounce on numerous pro-inflammatory mechanisms, define oxidative stress, genetic and epigenetic conditions, and have a hormone-sensitive action. [67–70].”.

Within the context of genetic conditions, it is recommended to be mentioned that PIK3CA-activating mutations in cooperation with loss of ARID1A expression seem to be necessary to initiating cancer development. In contrast, ARID1A inactivation alone is not sufficient for the oncogenic transformation of either the endometrium or ovarian surface epithelium.

Recommended reference: Samartzis EP, et al. Endometriosis-associated ovarian carcinomas: insights into pathogenesis, diagnostics, and therapeutic targets-a narrative review. Ann Transl Med. 2020;8(24):1712.

Author Response

REVIEWER N.3

this article, the authors analyze the association of the action of EDCs with EC from a molecular and clinical point of view. The manuscript is straightforward, well written, and concise and has clear result within the scope of a review article. Definitely deserves to be published and is a valuable contribution to the “International Journal of Molecular Sciences”. Some minor comments need to be addressed before publication.

Minor points:

  • “1. Introduction”, Lines 36-38:

“On the other hand, estrogen-independent endometrial neoplasms are associated with poor differentiation, arise later (menopause), have severe or clear cell differentiation, and are not associated with previous hyperplasia or dysplasia endometrium [4].”.

At that point, the authors should report whether there is an association between endometriosis and the development of endometrioid and clear cell carcinomas. An interesting paper reviewed cases of ovarian germ cell tumors (OGCT) in postmenopausal patients. Among 35 OGCT reported cases, endometrioid carcinoma was the most common epithelial component, and 6 of these events were associated with an endometriotic cyst. In terms of clear cell carcinoma, 1 out of 3 cases had a history of endometriosis as well.

  Recommended reference: Boussios S, et al. Malignant Ovarian Germ Cell Tumors in Postmenopausal Patients: The Royal Marsden Experience and Literature Review. Anticancer Res. 2015;35(12):6713-22.

Boussios S, Attygalle A, Hazell S, Moschetta M, McLachlan J, Okines A, Banerjee S. Malignant Ovarian Germ Cell Tumors in Postmenopausal Patients: The Royal Marsden Experience and Literature Review. Anticancer Res. 2015 Dec;35(12):6713-22. PMID: 26637887.

A: Thank you. The reference has been entered.

  • “3.1. EC AND BPA”, Lines 116-119:

“They report that in the EC, altered regulation of different MicroRNAs (miRNAs) are evident (reduction of the expression of miR-149 with adverse action on the cell cycle and increase of the expression of miR-107, which activates different mechanisms associated with the proliferation process cellular).”.

There are several studies where differential expression of miRNAs has been studied in either endometriosis or ovarian cancer. Several differentially expressed miRNA in endometriosis compared to ovarian cancer have been found, mainly linked with epithelial–mesenchymal transition. Two common miRNAs overexpressed in both diseases were miR-325 and miR-492. While the expression of miR-325 was upregulated in both diseases, this was more prominent in ovarian cancer, suggesting that miR-325 could have a role in the transition from endometriosis to ovarian cancer.

 Recommended reference: Braicu OL, et al. miRNA expression profiling in formalin-fixed paraffin-embedded endometriosis and ovarian cancer samples. Onco Targets Ther. 2017 Aug 28;10:4225-4238.

A: Thank you. The reference has been entered.

  • “4. Conclusion”, Lines 341-343:

“It has been possible for EDCs to pronounce on numerous pro-inflammatory mechanisms, define oxidative stress, genetic and epigenetic conditions, and have a hormone-sensitive action. [67–70].”.

Within the context of genetic conditions, it is recommended to be mentioned that PIK3CA-activating mutations in cooperation with loss of ARID1A expression seem to be necessary to initiating cancer development. In contrast, ARID1A inactivation alone is not sufficient for the oncogenic transformation of either the endometrium or ovarian surface epithelium.

Recommended reference: Samartzis EP, et al. Endometriosis-associated ovarian carcinomas: insights into pathogenesis, diagnostics, and therapeutic targets-a narrative review. Ann Transl Med. 2020;8(24):1712.

A: Thank you. The reference has been entered.
